# An Experimental Study on Field Spectral Measurements to Determine Appropriate Daily Time for Distinguishing Fractional Vegetation Cover

**Du Lyu [1,2]** , **Baoyuan Liu [1,3]**, **Xiaoping Zhang [1,3,]\***, **Xihua Yang [4]** , **Liang He [3]**, **Jie He [3]**, **Jinwei Guo [3]**, **Jufeng Wang [3]** and **Qi Cao [1]**

[1]  State Key Laboratory of Soil Erosion and Dryland Farming on the Loess Plateau, Institute of Soil and Water Conservation, Chinese Academy of Sciences and Ministry of Water Resources, Yangling 712100, China; lvdu18@mails.ucas.ac.cn (D.L.); baoyuan@bnu.edu.cn (B.L.); caoqi17@mails.ucas.edu.cn (Q.C.)

[2]  University of Chinese Academy of Sciences, Beijing 100049, China

[3]  Institute of Soil and Water Conservation, Northwest A&F University, Yangling 712100, China; 2013011805@nwafu.edu.cn (L.H.); xqhejie@nwafu.edu.cn (J.H.); hel@nwafu.edu.cn (J.G.); wangbaorong@nwafu.edu.cn (J.W.)

[4]  New South Wales Department of Planning, Industry and Environment, Parramatta, NSW 2124, Australia; xihua.yang@environment.nsw.gov.au

\*  Correspondence: zhangxp@ms.iswc.ac.cn

**Abstract:** Remote sensing technology has been widely used to estimate fractional vegetation cover (FVC) at global and regional scales. Accurate and consistent field spectral measurements are required to develop and validate spectral indices for FVC estimation. However, there are rarely any experimental studies to determine the appropriate times for field spectral measurements, and the existing guidelines or references are rather general or inconsistent, it is still not agreed upon and detailed experiments are missing for a local research. In this experiment, five groundcover objects were measured continuously from 07:30 a.m. to 17:30 p.m. local time in three consecutive sunny days using a portable spectrometer. The coefficients of variation (CV) were applied to investigate the reflectance variation at wavelengths corresponding to MODIS satellite channels and the derived spectral indices used to estimate FVC, including photosynthetic vegetation (PV) and non-photosynthetic vegetation (NPV). The results reveal little variation in the reflectance measured between 10:00 a.m. and 16:00 p.m., with CV values generally less than 10%. The CV values of FVC spectral indices for estimating PV, NPV and bare soil (BS) are generally less than 3%. While more experiments are yet to be carried out at different locations and in different seasons, the findings so far imply that the in situ spectrum measured between 9:00 a.m. and 17:00 p.m. local time would be useful to discriminate FVC objects and validate satellite estimates-based indices using visible, near-infrared and shortwave infrared channels.

**Keywords:** fractional vegetation cover; field spectral measurement; spectral indices; appropriate measurement time

## 1. Introduction

As an integral part of ecosystems, vegetation, including photosynthetic (PV, green leaves) and non-photosynthetic (NPV, aboveground dead biomass, litter and wood), plays an important role in climate regulation, geochemical cycle, and soil and water conservation [1–3]. Fractional vegetation cover (FVC) is the ratio of vertical projection area of vegetation to total ground area, which is usually used to evaluate the degree of land degradation [4] and the function of soil and water conservation [5,6], and widely used in various soil erosion prediction models, such as USLE [7], RUSLE [8] and CSLE

model [9]. Therefore, quantitative estimation of the fractional cover of PV ($f_{PV}$) and NPV ($f_{NPV}$) is critical for the accurate assessment of its function in the terrestrial system. As the basis of remote sensing technology, ground object spectrum or reflectance is used to detect the physical characteristics of the ground targets from the space. Field spectral measurements or ground truthing is often required to build the relationship between the ground targets and the sensor or calibration. In vegetation studies, the field measurement of PV, NPV, and BS (bare soil) can not only help understand the differences of spectral characteristics among them, but also provide means of validation and development of suitable spectral indices for estimation of $f_{PV}$ and $f_{NPV}$ [10].

However, it is found that there is no consistent description of the appropriate daily time for field spectral measurement. According to the national standard in China, the suitable time for field spectral measurement is 10:00 a.m. to 15:00 p.m. [11]. Guerschman et al. described the time for spectrum acquisition of PV, NPV and BS for sparse grassland in Australia as 10:00 a.m.–16:00 p.m. [12]. Cao et al. collected the spectral data in Xilingol grassland of China from 10:00 a.m. to 14:00 p.m. [13]. It can be seen that the daily time for field spectral measurements of PV, NPV and BS was inconsistent probably due to the different solar radiation energy resulting from different geographical locations.

In reality, field travel also makes it difficult to collect field spectral data at the desirable time. In addition, the satellites overpass times are often fixed at any locations, for example, MODIS Terra and Landsat-8 at about 10:30 a.m. and MODIS Aqua at 1:30 p.m. Though it is ideal to perform the field spectral measurements exactly at the same times as satellites overpass, this is not always possible.

It is well known that leaves of PV have characteristics exclusively on the spectrum of VIR-NIR (400–1100 nm) which make it easily distinguishable from NPV and BS. Based on this characterization, the normalized differences vegetation index (NDVI) is therefore designed to identify PV and green canopy, and is one of the most powerful vegetation indices used in regional vegetation detection whatever remote sensors are used, such as Moderate Resolution Imaging Spectroradiometer (MODIS), Landsat-TM and Hyperion [14].

It is found that the different reflectivity characteristics exit NPV and BS at the shortwave infrared band (SWIR, 1100–2400 nm). Therefore, it was possible to design the suitable indices to identify NPV from BS using the spectral characteristics of SWIR. The hyperspectral cellulose absorption index (CAI), based on Hyperion satellite images, provides the best estimation of the fractional cover of crop residues and BS using the absorption properties of NPV cellulose at 2100 nm [15]. However, hyperspectral images are difficult to obtain and not suitable for regional analysis. Other researchers proposed some NPV indices based on multispectral Landsat TM images, such as the normalized difference index (NDI) [16], normalized difference senescent vegetation index (NDSVI) [17]. The disadvantages are that these indices can only distinguish NPV or BS from the PV background, and are not applicable when the three components of PV, NPV and BS coexist. The MODIS-based STI (Soil Tillage Index) and SWIR32 index (shortwave infrared ratio), are both based on the spectral differences in band 7 and band 6 of MODIS multispectral image. While the latter index is proved to be more effective than the former to distinguish NPV from BS [18]. With its advantages of larger-area, high temporal resolution, and low-cost monitoring, MODIS data holds greater potential to identify regional FVC. Therefore, NDVI and SWIR32 indices extracted from MODIS bands are supposed to be feasible for estimating $f_{PV}$ and $f_{NPV}$ in our experiment.

The four common questions are: (1) will the reflectivity of the targeted objects change over time and by how much? (2) Will the spectral indices used to distinguish PV, NPV and BS change with different acquisition times? (3) Will the time difference between the satellite overpass and the ground measurement affect PV and NPV estimation? (4) What is the appropriate time for field spectrum measurement and the indices for discriminating FVC? In order to answer these questions, this study designed an experiment and set up five FVC objects including one green wheat for PV, three types of withered leaves of natural vegetation communities for NPV types and a bare soil for BS to measure their reflectance curves outside on sunny days from morning to evening at one-hour intervals. The spectral reflectance values of the targets at different times and the derived spectral indices were investigated to

detect their variance characteristics. It is expected to be helpful in selecting appropriate daily time for field spectral measurement, and providing a means of validation for corresponding remotes sensing images in estimating vegetation coverage.

## 2. Materials and Method

### 2.1. Experimental Design

This study collected the withered leaves of *Quercus* Linn. in the Qinling Mountains (107°33′19″ E; 34°4′9″ N; 1497 m a.s.l.) of the Loess Plateau as NPV1, where it was the main constructive tree species, *Acer ginnala* Maxim. in the Ziwuling (109°1′7″ E; 35°43′2″ N; 1171 m a.s.l.) as NPV2, where it was a dominant species, and *Stipa bungeana* Trin. as NPV3 of the typical grassland (109°11′33″ E; 37°49′47″ N; 1244 m a.s.l.) in Hengshan County, where it was a dominant grass species. The surface soil of the farmland was collected as bare soil in Yangling (108°4′23″ E; 34°17′30″ N; 520 m a.s.l.) (an agricultural area in the Loess Plateau), and the particles were passed through a 2-mm sieve and measured as BS sample. The specific soil type around the area is called Lou soil with 2.5% sand, 67.2% silt, 30.3% clay, and the content of organic matter is about 17 g/kg [19].

The experimental boxes were designed with the size of 40 × 40 × 20 cm (length × width × height) as shown in Figure 1. Three sample boxes contained the three types of NPV samples, one for the bare soil sample, and another for the PV sample (wheat, *Triticum aestivum* L.). The wheat was planted inside the box with the density of 4–5 seeds per cm$^2$, when it grew up to cover 100% of the sample box about two weeks and measured as the PV sample.

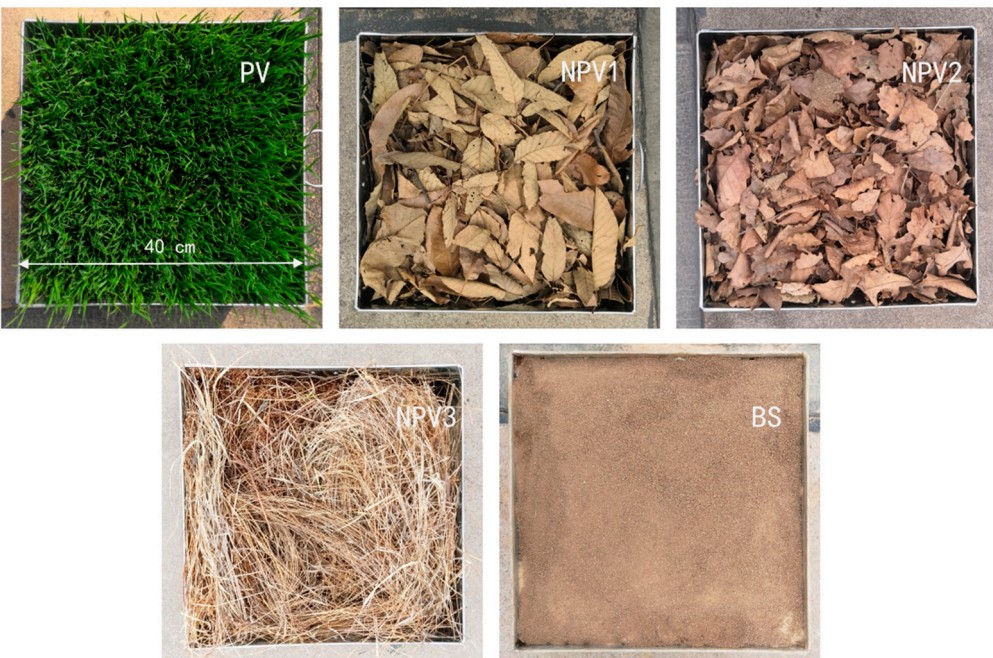

**Figure 1.** The experimental samples of five fractional vegetation cover (FVC) objects (photosynthetic vegetation (PV), green leaves of wheat, *Triticum aestivum* L., 16 cm high on average; non- photosynthetic vegetation (NPV)1, withered leaves of *Quercus* Linn., 12 cm long and 5 cm wide on average; NPV2, withered leaves of *Acer ginnala* Maxim., 7 cm long and 4.5 cm wide on average; NPV3, withered leaves of *Stipa bungeana* Trin., 50 cm long and 0.6 cm wide on average; bare soil (BS), Lou soil).

To reduce the effect of the water content of the leaves surface and soil on the spectral reflectance, the leaves and soil were left in a dry, cool place for a week before being tested. Measurements were conducted on December 8, 9 and 10, 2019. The sky in these three days was clear and sunny, and wind speed was less than 4.5 m/s. The observation was carried out from 7:30 a.m. to 17:30 p.m. local time,

the hourly weather information is shown in Table 1. The measurement site was located in Yangling District, Shaanxi Province (108°4′33″ E; 34°16′33″ N) at a 482 m a.s.l.

**Table 1.** Temperature and Humidity Record at Twelve Measuring Time Points.

| Project | Date | 7:30 | 8:00 | 9:00 | 10:00 | 11:00 | 12:00 | 13:00 | 14:00 | 15:00 | 16:00 | 17:00 | 17:30 |
|---|---|---|---|---|---|---|---|---|---|---|---|---|---|
| | 8 December | −3 | −4 | −2 | 0 | 3 | 7 | 9 | 10 | 11 | 11 | 11 | 10 |
| Temperature/°C | 9 December | −4 | −4 | −1 | 3 | 8 | 11 | 12 | 12 | 13 | 15 | 13 | 12 |
| | 10 December | 0 | 0 | 1 | 7 | 11 | 13 | 14 | 15 | 15 | 15 | 14 | 13 |
| | 8 December | 90 | 90 | 91 | 68 | 50 | 48 | 48 | 47 | 44 | 45 | 47 | 52 |
| Humidity/% | 9 December | 86 | 87 | 84 | 68 | 48 | 38 | 35 | 32 | 30 | 27 | 26 | 26 |
| | 10 December | 69 | 65 | 62 | 40 | 24 | 19 | 16 | 17 | 17 | 17 | 18 | 19 |
| | 8 December | | | | | | 07:36 and 17:41 | | | | | | |
| Sunrise and Sunset | 9 December | | | | | | 07:37 and 17:41 | | | | | | |
| | 10 December | | | | | | 07:38 and 17:42 | | | | | | |

Spectral measurements of PV, NPV and BS samples were performed by SVC HR-1024i portable hyper-spectrometer (USA) with a spectral range of 350–2500 nm, the spectral resolution was 3.5 nm for 350–1000 nm, 9.5 nm for 1000–1850 nm and 6.5 nm for 1850–2500 nm. The field of view of the probe was 25°.

To reduce the impact of shelter, the probe was kept vertically downward with 50 cm above the sample center. The target and 95% standard whiteboard were measured in turn. Each object was measured 4 times at each time point, and 3 consecutive spectral curves were measured each time for a total of 12 curves as replicates per day, and the average of a total of 36 repetitions over three days was obtained as the reflection spectrum for each object at that time point.

*2.2. Bands Identification and Spectral Indices Calculation*

All ground objects have the ability to reflect electromagnetic radiation, and most of the electromagnetic radiation energy reflected by objects on the surface of the earth comes from solar energy. Reflected radiation energy of the object as a percentage of the total radiation energy, known as reflectivity, is dimensionless. The changing law of the reflectance of an object with the incident wavelength is called the reflection spectrum of the object, which is commonly expressed as the reflection spectrum curve [20].

The reflectance curves of PV, NPV and BS from our experiment are shown in Figure 2 The horizontal axis represents the wavelength, and the vertical axis represents reflectance. These wavelengths heavily affected by the atmosphere and the water vapor absorption were excluded, and retaining wavelengths are 350–1300, 1450–1750, and 2000–2300 nm. The spectral characteristics of PV, NPV and BS were here analyzed using the measured spectral curves of the five objects at 12:00 p.m. (noon) as an example (Figure 2).

Due to the influence of chlorophyll, PV had the typical spectral characteristics of green vegetation. At the green band (560 nm) of VIR (visible band, 400–700 nm), there was a small reflection peak, and near the red band (670 nm), there was an absorption valley. In addition, the reflectivity of PV showed a significant high reflectivity in NIR (near infrared, 700–1100 nm). Therefore, PV could be easily distinguished from NPV and BS in VIR-NIR (400–1100 nm). However, NPV and BS not only had no special spectral characteristics in VIR-NIR, but also had similar reflectivity curves, thus it was impossible to distinguish NPV and BS by using VIR-NIR.

At the wavelength of SWIR (shortwave infrared, 1100–2400 nm), the reflectivity of NPV and BS was quite different. NPV had obvious reflection peak at 1700 nm, and had diagnostic absorption characteristics affected by cellulose near 2100 nm at the same time. In addition, BS had another absorption characteristic at 2200 nm mainly referring to the lattice of clay minerals. Therefore, it was possible to estimate $f_{NPV}$ using the spectral characteristics of SWIR.

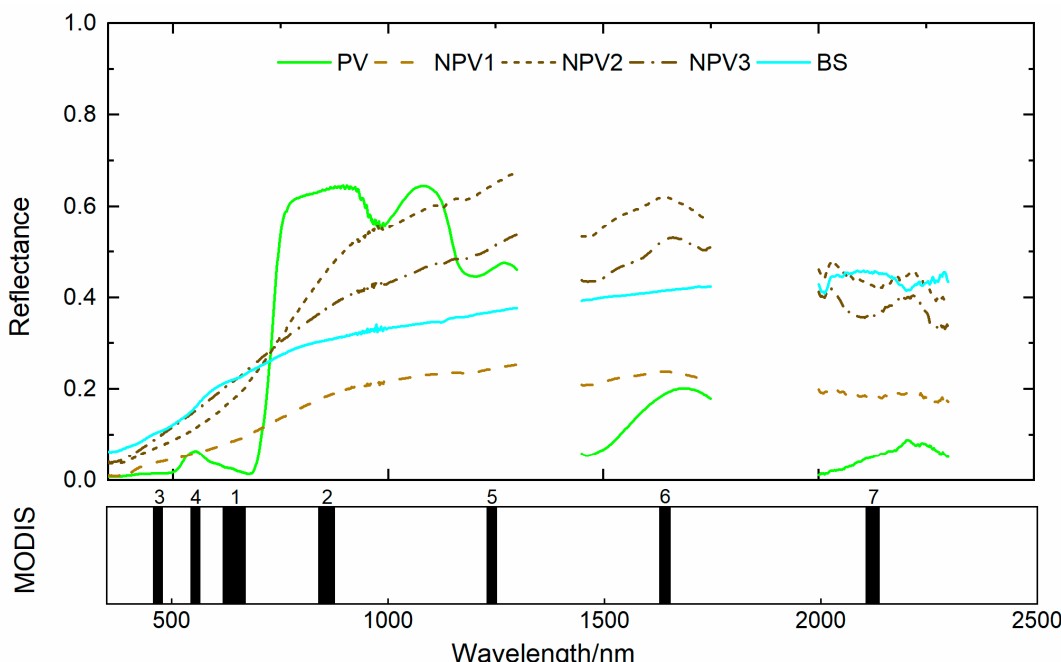

**Figure 2.** Field spectral reflectance of PV, NPV and BS measured at 12:00 p.m. noon at corresponding MODIS channels. (Specific information about PV, NPV1, NPV2, NPV3 and BS is provided in the notes of Figure 1. Figure 2 also provides the 1–7 channels of the MODIS sensor, with the black bandings representing the band ranges covered by each channel.)

This study selected the most widely used NDVI index for estimating $f_{PV}$, as well as the SWIR32 for estimating $f_{NPV}$ based on MODIS multispectral image. Both two indices were calculated after resampling the measured data according to band range of the MODIS sensor (Figure 2). Following the MODIS channels, the combination of absorption band 1 (red band, 640–670 nm) and high reflectance band 2 (NIR, 700–800 nm) for PV could be used to derive NDVI. The reflectance peak of band 6 (SWIR2, 1628–1652 nm) and absorption valley of band 7 (SWIR3, 2105–2135 nm) could be designed (SWIR32) for NPV estimation.

In our experiment, the 36 reflectance curves of PV at each time point were used to calculate NDVI following Equation (1). The 36 reflectance curves of the four objects of NPV1, NPV2, NPV3 and BS at each time point were used to calculate SWIR32 following Equation (2). The index value for each time point was calculated by averaging 36 replications. The calculation formula of each indices was referenced in Guerschman et al. [21]. MODIS$_X$ indicated the band number of the MODIS satellite sensor.

$$NDVI = (MODIS2 - MODIS1)/(MODIS2 + MODIS1) \tag{1}$$

$$SWIR32 = MODIS7/MODIS6 \tag{2}$$

*2.3. Variation Test*

In order to determine the appropriate daily time for field spectral acquisition, the variation of reflectance values over time of band 1, 2, 6 and 7 and the derived indices (NDVI, SWIR32) were investigated and compared over the measurement period.

First, the coefficients of variation (CV %) of the 36 repetitions of both reflectance values and spectral indices at each time point were investigated in this study. Then, this study divided the 12 time points during the period 7:30 a.m.–17:30 p.m. into 55 time periods with 3 or more consecutive time points, and then determined the most appropriate daily time period based on the CV analysis of the averaged reflectance or the averaged indices at each time period. A CV of less than 10% was considered as acceptable, as suggested by Duggin [22].

Finally, One-way ANOVA was applied to further analyze the significance differences of the spectral indices over 12 time points from 7:30 a.m. to 17:30 p.m. by using 36 repeating values at each time point, taking 95% as confidence level.

## 3. Results

### 3.1. Refelctance Variation of Characteristic Bands over Time

This study began with a qualitative analysis of the reflectance of the characteristic bands that can best distinguish PV, NPV and BS over time. The measured spectral data were resampled according to the corresponded channels 1, 2, 6, 7 of MODIS sensor. As shown in Figure 3a, the two gray lines show the reflectance of PV in band 1 and 2 corresponding to MODIS. It can be seen that the reflectance of band 1 is small and varies little with time. The reflectance of band 2 varies widely and is relatively stable between 11:00 a.m. and 16:00 p.m. Similarly, the gray lines in Figure 3b–e show the temporal change of reflectance for NPV1–3 and BS in band 6 and band 7 corresponding to the MODIS, respectively. It can be seen that the band 6 and 7 have a large dispersion at the time points near sunrise and sunset at 7:00 a.m., 8:00 a.m. and 17:30 p.m., and even show error values greater than one.

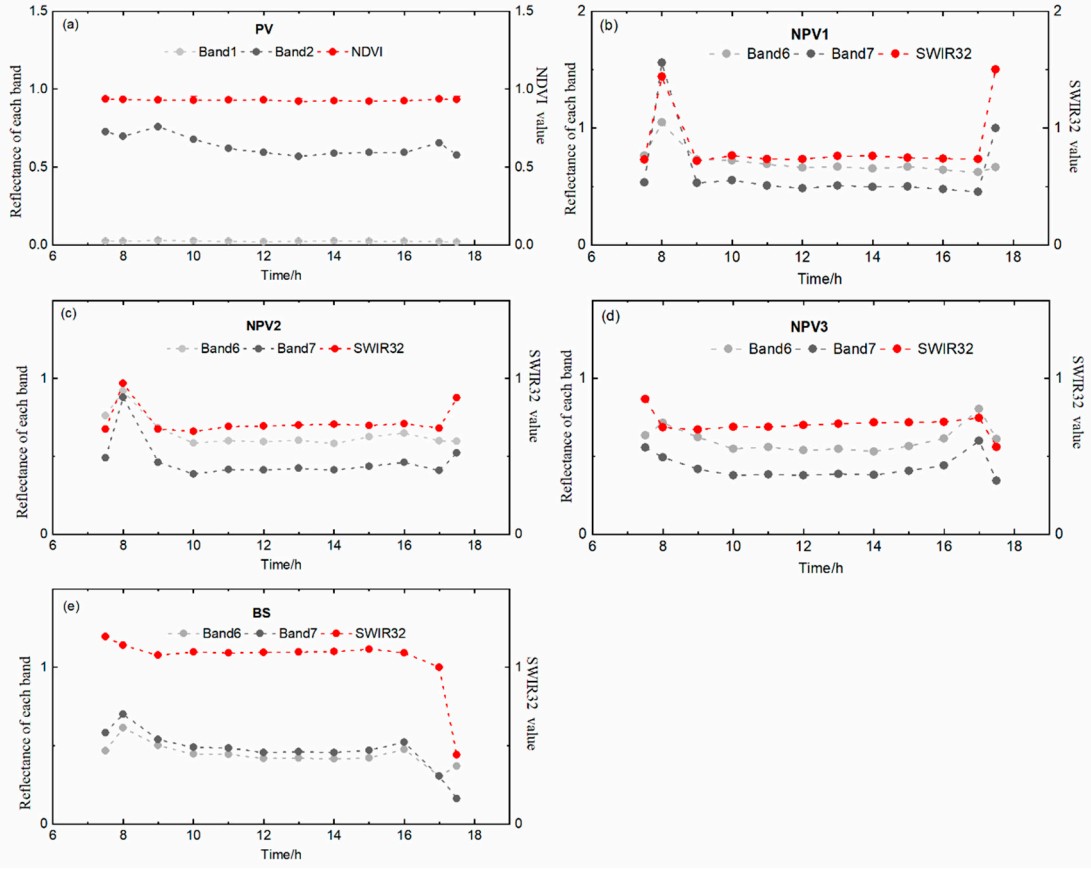

**Figure 3.** Change of the averaged reflectance values over time at the four MODIS bands and the associated spectral indices for the five FVC objects. ((a–e) represent the five FVC objects respectively. The specific information about PV, NPV1, NPV2, NPV3 and BS is consistent with Figure 1. The band(X) represents the reflectance of a band obtained by resampling field spectra according to the band range of MODIS sensor.).

In order to further verify the degree of variation over time of these reflectance values and spectral indices as shown in Figure 3, the mathematical analysis was used to calculate the CV. As described in Section 2.3, the first method used 36 spectral curves at each time point to extract the reflectance of the

characteristic bands corresponding to MODIS, and 36 replicate values of each band were used as a sample to calculate the CV; the results are shown in Table 2.

**Table 2.** Coefficient of Variation of Reflectance in the MODIS Channels and Spectral Indexes.

| CV/% | 7:30 | 8:00 | 9:00 | 10:00 | 11:00 | 12:00 | 13:00 | 14:00 | 15:00 | 16:00 | 17:00 | 17:30 |
|---|---|---|---|---|---|---|---|---|---|---|---|---|
| PV-band1 | 24.05 | 19.64 | 40.23 | 47.16 | 29.78 | 7.66 | 13.01 | 14.25 | 14.77 | 16.87 | 12.53 | 19.24 |
| PV-band2 | 19.87 | 11.12 | 21.06 | 9.69 | 16.63 | 8.27 | 8.84 | 6.14 | 7.30 | 12.91 | 20.41 | 18.88 |
| PV-NDVI | 1.10 | 0.87 | 1.62 | 2.68 | 1.18 | 0.37 | 1.67 | 0.54 | 0.86 | 0.54 | 0.87 | 1.85 |
| NPV1-band6 | 15.42 | 32.82 | 22.63 | 14.59 | 9.16 | 5.40 | 11.69 | 15.56 | 7.08 | 14.77 | 17.68 | 12.55 |
| NPV1-band7 | 327.34 | 86.17 | 28.81 | 17.39 | 13.81 | 8.77 | 14.38 | 16.97 | 9.88 | 19.45 | 17.88 | 189.04 |
| NPV1—SWIR32 | 114.50 | 90.98 | 10.40 | 5.58 | 6.29 | 0.06 | 5.88 | 6.51 | 5.85 | 6.56 | 8.51 | 89.63 |
| NPV2-band6 | 15.56 | 34.08 | 17.65 | 10.99 | 7.78 | 4.83 | 7.62 | 5.41 | 6.27 | 7.20 | 13.13 | 13.98 |
| NPV2-band7 | 299.12 | 58.51 | 21.48 | 12.95 | 8.25 | 5.41 | 7.84 | 5.60 | 8.65 | 10.27 | 18.94 | 287.76 |
| NPV2-SWIR32 | 153.57 | 37.25 | 6.39 | 2.51 | 3.58 | 2.83 | 1.76 | 2.85 | 3.82 | 4.00 | 7.70 | 171.32 |
| NPV3-band6 | 18.24 | 17.48 | 18.87 | 9.54 | 7.14 | 6.24 | 3.66 | 4.24 | 7.10 | 9.10 | 46.14 | 21.05 |
| NPV3-band7 | 240.08 | 23.47 | 19.39 | 8.76 | 6.95 | 6.44 | 5.16 | 5.15 | 8.13 | 11.87 | 47.65 | 442.89 |
| NPV3-SWIR32 | 103.76 | 9.47 | 3.61 | 2.38 | 1.98 | 2.54 | 2.82 | 1.61 | 2.34 | 3.19 | 7.01 | 193.30 |
| BS-band6 | 16.29 | 25.63 | 14.04 | 6.82 | 9.14 | 6.54 | 6.27 | 6.82 | 10.17 | 7.17 | 39.14 | 19.90 |
| BS-band7 | 179.04 | 28.49 | 12.98 | 6.03 | 9.43 | 6.47 | 5.37 | 6.97 | 10.79 | 8.68 | 46.88 | 938.47 |
| BS-SWIR32 | 120.39 | 3.45 | 3.86 | 1.37 | 1.33 | 1.33 | 1.53 | 1.29 | 1.49 | 3.98 | 12.23 | 410.88 |

Note: CV, coefficient of variation (standard deviation/mean). (The specific information about PV, NPV1, NPV2, NPV3 and BS is consistent with Figure 1. The band(X) represents the reflectance of a band obtained by resampling field spectra according to the band range of MODIS sensor.).

It reveals that the reflectivity of band 1 from PV varies greatly among 36 repetitions, CV of 36 repetitions of band 2 is around 10% in the period from 10:00 a.m. to 16:00 p.m., the average CV over the time period is 9.97%. The CV of band 6 and band 7 for NPV1 is around 10% at 10:00 a.m.–16:00 p.m., with the averaged CV of 11.17% and 14.37%, respectively. The CV of bands 6 and 7 for NPV2 is beneath 10% from 10:00 a.m. to 16:00 p.m., with the averaged CV of 7.15% and 8.42%, respectively. The CV of bands 6 and 7 for NPV3 is less than 10% from 10:00 a.m. to 16:00 p.m., with the averaged CV of 6.71% and 7.49%, respectively. The CV of BS's band 6 and 7 is less than 10% between 9:00 a.m. and 16:00 p.m. with the averaged CV of 8.37% and 8.34%, respectively.

In general, the reflectance of these FVC objects was relatively stable from 10:00 a.m. to 16:00 p.m. due to the CV of less than 10%, except for band 6 and 7 of NPV1 and band 1 of PV. In particular, the reflectance of band 6 and 7 of BS was stable even from 9:00 a.m. to 16:00 p.m.

The second method was used here to check the variability by selecting different samples to calculate the CV. Figure 4 shows 15 plots which indicate the CV about characteristic bands of PV, NPV1–3 and BS, as well as the calculated spectral indices (NDVI or SWIR32) for each time period. The vertical axis represents the start time point and the horizontal axis is the end time point. Each grid in the triangular matrix represents the CV value of the period from its corresponding start time to end time, and the total 55 time periods are obtained in each plot. The lightest gray grid in the figure represents the time period in which the CV is in the acceptable 0–10% range.

According to the CV of reflectance values over 55 time periods (Figure 4), it is found that reflectance of band 1 of PV was relatively stable from 10:00 a.m. to 17:00 p.m. with the CV of 7.57%. For all the time slices within the frame from 10:00 a.m. to 17:00 p.m., such as 10:00 a.m.–12:00 p.m., 10:00 a.m.–13:00 p.m, . . . , 15:00–17:00 p.m., the corresponding CV values are 9.35%, 8.47%, and 5.79%, respectively, which are all within the acceptable range (0–10%). It is reasonable to conclude that in the time period of 10:00 a.m.–17:00 p.m., the band reflectivity is relatively stable. Similarly, Band 2 of PV was stable from 10:00 a.m. to 17:30 p.m., with a CV of 6.04%, for all intra-periods of 10:00 a.m.–12:00 p.m., 10:00 a.m.–13:00 p.m., . . . , 16:00–17:30 p.m. the CVs are all less than 10%.

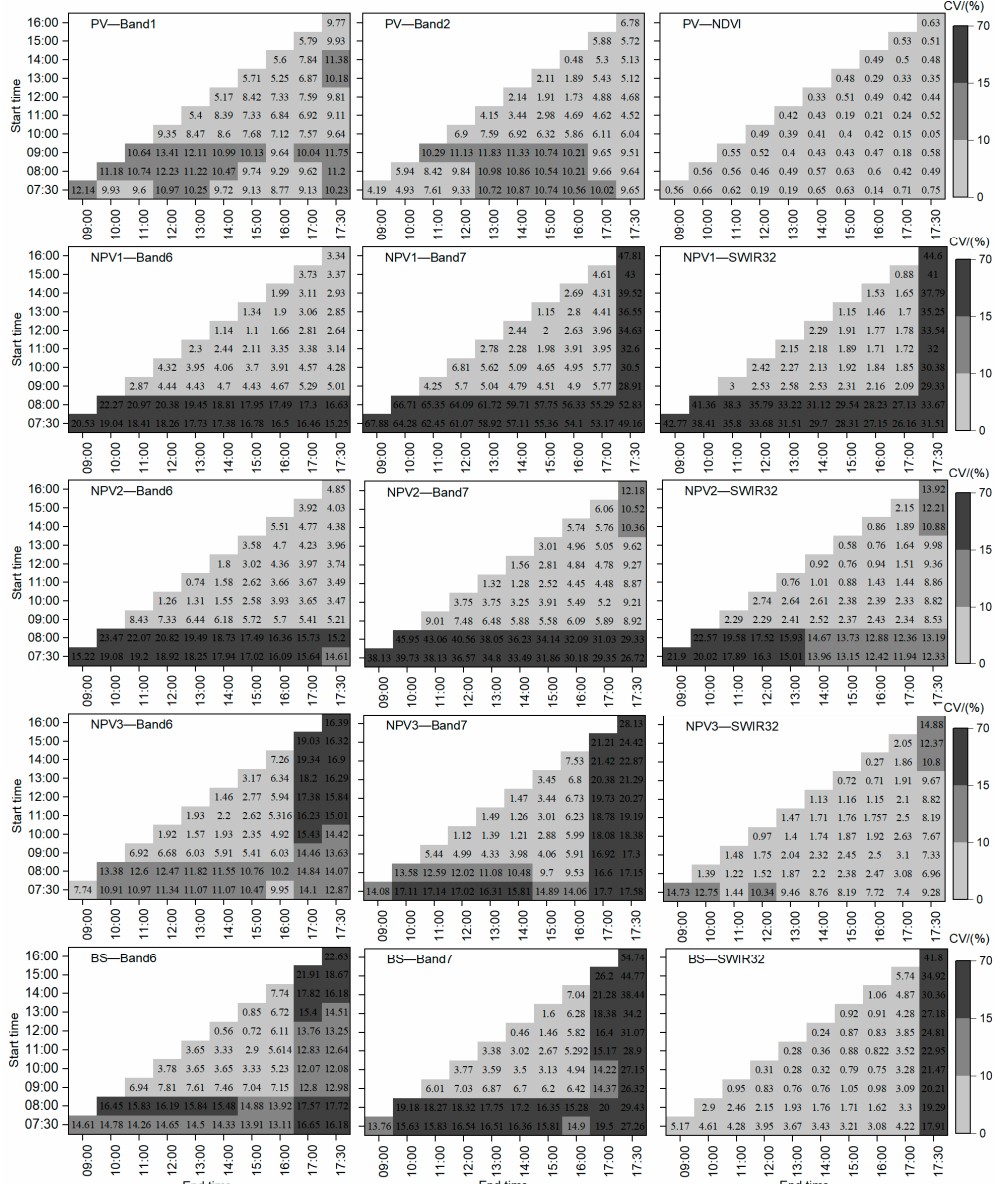

**Figure 4.** Coefficient of variation (CV) (%) of reflectance and spectral indices at different time periods. (The specific information about PV, NPV1, NPV2, NPV3 and BS is consistent with Figure 1. The band(X) represents the reflectance of a band obtained by resampling field spectrum according to the band range of MODIS sensor.).

Moreover, band 6 and 7 of NPV1 kept stable at 9:00 a.m.–17:00 p.m. and 9:00 a.m.–17:30 p.m. with the CV of 5.01% and 5.77%, respectively. Band 6 and 7 of NPV2 were stable between 9:00 a.m. and 17:30 p.m. and 9:00 a.m.–17:00 p.m. with CV of 5.21% and 5.89%, respectively. Band 6 and 7 of NPV3 were stable from 9:00 a.m. to 16:00 p.m. with CV of 6.03% and 5.91%, respectively. Similarly, band 6 and 7 of BS were also stable from 9:00 a.m. to 16:00 p.m. with CV of 7.15% and 6.42%, respectively.

From Table 2 and Figure 4, the appropriate daily time identified by two methods was different, and the set of time by the first method was slightly strict. In general, the CV of reflectance in the period of 10:00 a.m.–16:00 p.m. was acceptable for all the five FVC objects.

## 3.2. Effectiveness of Spectral Indices for Distinguishing FVC Objects

Based on experimental reflectance, the effectiveness of NDVI and SWIR32 to differentiate the PV, NPV, and BS was tested following Equations (1) and (2).

The hourly variation of NDVI among the five FVC objects was shown in Figure 5a; Figure 5b represents the change in SWIR32 index hourly for five objects. From Figure 5a, it can be seen that the NDVI index of PV was significantly high, with a range of 0.921–0.935 over time in the period of 7:30 a.m.–17:30 p.m. The NDVI index of NPV fluctuated from 0.242 to 0.447, and the NDVI index of BS changed around 0.104 to 0.177 over the time. The NDVI values of PV could be much easier to differentiate than that of NPV and BS which made it effective to detect green vegetation from other objects.

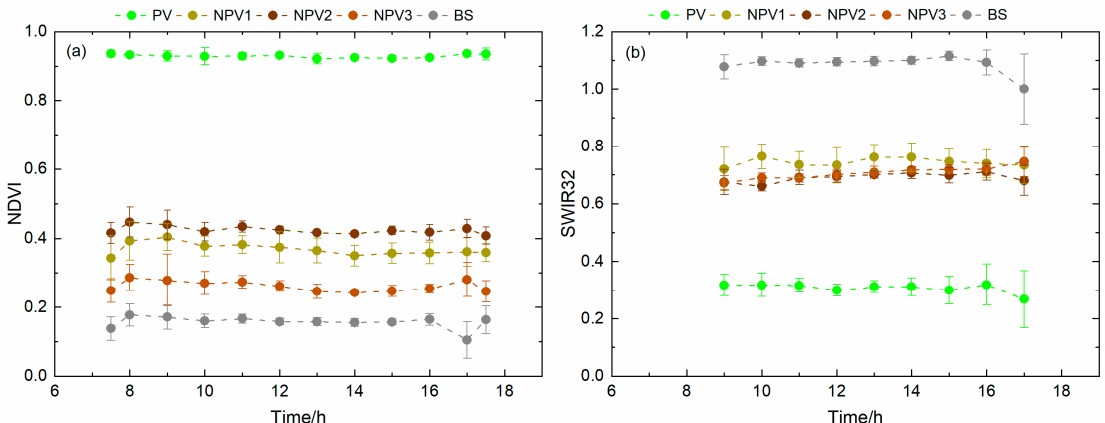

**Figure 5.** The hourly variation of NDVI and SWIR32 among the five FVC objects. (The specific information about PV, NPV1, NPV2, NPV3 and BS is consistent with Figure 1.).

Since the reflectance of 6 and 7 bands of five kinds of FVC objects had abnormal values at 7:30 a.m., 8:00 a.m. and 17:30 p.m., the SWIR32 index of these 3 h were excluded in the Figure 5b. Figure 5b shows a very clear pattern of the distribution of SWIR32 of PV, NPV and BS. The SWIR32 index of PV, NPV and BS were 0.267–0.317, 0.661–0.764, and 1.0–1.116, respectively. It is evident that SWIR32 is an effective index to distinguish NPV from BS.

### 3.3. CV of the Spectral Indices over Time

This study used three methods to evaluate the variation of the spectral indices over time, including two kinds of CV detection method and ANOVA.

The first method is consistent with the method used to calculate the CV value of reflectance in Table 2. The CV of NDVI and SWIR32 between different hours was calculated using their 36 repetitions on each time point, and the results are shown in Table 2. Compared with the great variation of band 1 and 2 of PV, the NDVI of PV was much more stable, and the CV was no more than 2% in all 36 replicates at all-time points from 7:30 a.m. to 17:30 p.m. (Table 2). Similarly, the CV of 36 repetitions of both SWIR32 index of NPVs and BS was about 10% from 9:00 a.m. to 17:00 p.m., which was more stable than reflectance of their characteristic bands with time.

The results of the second method are shown in Figure 4. In addition to the CV of the reflectance analyzed in Section 3.1, Figure 4 also shows that the CV of FVC indices (NDVI and SWIR32) over the 55 time periods is stable from 7:30 a.m. to 17:00 p.m. for NDVI of PV with the value of 0.75%, the SWIR32 of NPV1 and NPV2 remained stable from 9:00 a.m. to 17:00 p.m. with the CV of 2.09% and 2.34%, respectively. For NPV3, the SWIR32 was stable from 8:00 a.m. to 17:00 p.m. with CV of 3.08%. SWIR32 of BS was stable from 7:30 a.m. to 17:00 p.m. with CV of 4.22%.

From Table 2 and Figure 4, in general, compared with the absolute reflectance value of the ground objects, the derived spectral indices, NDVI and SWIR32, could remain largely stable in the time point from 9:00 a.m. to 17:00 p.m. During this period, the CVs are 0.18% for NDVI, 2.09%, 2.34%, 3.1% and 3.09% for NPV1, NPV2, NPV3 and BS, respectively.

### 3.4. Significance Test for Spectral Indices over Time

One-way ANOVA was carried out to further test the variation of each index which was used to estimate $f_{PV}$ or $f_{NPV}$ over the time (Figure 5). The results showed that, the variation of the PV-NDVI between 9:00 a.m. and 17:00 p.m. was not significant ($p > 0.05$). The difference of SWIR32 for NPV1, NPV2, NPV3 and BS was all not significant from 9:00 a.m. to 17:00 p.m. ($p > 0.05$).

Table 3 lists the appropriate time periods (I and II) identified by the CV of less than 10% as the results of Table 2 and Figure 4, and the suitable time period (III) obtained by One-way ANOVA (Figure 5). Combination of three methods showed that both NDVI and SWIR32 were stable from 9:00 a.m. to 17:00 p.m., indicating that measuring spectra for this time period would satisfy the requirement of vegetation cover estimation using remote sensing technology. It is also implied that the derived spectral indices obtained in this period can be comparable to those extracted from MODIS Terra images.

**Table 3.** Appropriate Daily Time Determined by Three Methods Based on Spectral Indices.

| Index | Appropriate Time I | Appropriate Time II | Appropriate Time III |
|---|---|---|---|
| PV-NDVI | 7:30–17:30 | 7:30–17:30 | 9:00–17:00 |
| NPV1-SWIR32 | 9:00–17:00 | 9:00–17:00 | 9:00–17:00 |
| NPV2-SWIR32 | 9:00–17:00 | 9:00–17:00 | 9:00–17:00 |
| NPV3-SWIR32 | 8:00–17:00 | 8:00–17:00 | 9:00–17:00 |
| BS-SWIR32 | 8:00–17:00 | 7:30–17:00 | 9:00–17:00 |

Note: The specific information about PV, NPV1, NPV2, NPV3 and BS is consistent with Figure 1. The appropriate time I and II mean the daily time periods detected by the mathematical methods defined in Section 2.3, the appropriate time III means the daily time period from One-way ANOVA test in this section.

In addition, different spectral indices seemed to have different requirements for spectrum measurement time. The NDVI index of PV performed more stably, and the time range suitable for spectrum acquisition was longer, while the SWIR32 index of NPV and BS was shorter. Although the range of characteristic bands was the same, the SWIR32 of NPV was still stricter than that of BS in terms of time.

In summary, 10:00 a.m.–16:00 p.m. was a strict time period in consideration of the variation of reflectivity of each characteristic band. Between 9:00 a.m. and 17:00 p.m. was also acceptable when the derived spectral indices were only considered over time.

## 4. Discussion

### 4.1. Variation of Reflectance over Time

Reflectance is influenced by ancillary factors including atmospheric scattering and absorption, topography, slope and aspect, solar zenith angle, and even earth–sun distance [23]. Jackson et al. reported that the percentage error in the reference panel irradiance increased with the increase in the solar zenith angle [24]. Chang et al. believed that because atmospheric conditions were variable, the calculation of reflectivity would introduce errors without measuring the irradiance of both the target and the reference plate, and the error would increase with the increase in cloud cover [25]. Kimes et al. (1983) reported errors in spectral radiation due to nearby objects in field studies: people holding sensors, backgrounds, buildings, or trees [26]. Some studies have pointed out that the quality of the reference panels at the time of measurement can also introduce errors, as it is impossible to construct a perfect Lambert surface [27,28].

Duggin compared two different methods of measuring reflectance. One is a sequential measurement and the other is a simultaneous measurement, showing that sequential measurements introduce more error in the reflectance calculations [29]. These errors can be minimized by creating similar conditions in the same fields at the same time [25,30].

In order to obtain the reflectance of the target, measuring the irradiance of the standard reflectors and the target is needed. The reflectance of the target is the irradiance ratio of the target and the standard reflectors. This is based on the assumption that the intensity and distribution of irradiance is invariant during readings of the target and standard reflectors [31]. Under the natural light, the radiation received by the ground objects mainly comes from the direct sunlight in the visible and infrared band. When the sun rises or sets at a very low solar altitude, the energy of direct sunlight is weak and influenced easily by temperature and humidity, which will attenuate the radiation intensity through the comprehensive effect of reflection, absorption and scattering [32]. The resulting measurement error in radiation intensity between the standard reflector and the target is supposed to be the dominant reason why the CV of reflectance of objects at early morning and at dusk was very large for the 36 repetitions at the certain time point or in comparison with the 12 time points.

Additionally, the characteristics of the object itself and the uniformity of its surface can also have an impact on the results. In our experiment, the degree of dispersion of reflectance with time fluctuations varies for different wavelengths of different objects. Compared with that of NPV1 and NPV2, the CV between 36 repetitions of reflectance in the band 6 and band 7 of NPV3 and BS was smaller. This was probably related to the surface homogeneity of different objects which will affect the radiation intensity. NPV1 and NPV2 were large broad leaves, and the shadows generated by leaf folding would have an impact on the measurements, while NPV3 as tiny grass leaves and BS as a sifted uniform soil, were evenly distributed within the field of view.

Due to the influence of additional reflectivity, compared with single leaf, multiple leaves can produce higher reflectivity in the NIR band of the spectrum [33]. In our experiment, although the NDVI of PV was very stable, the reflectance in band 2 (NIR) of PV had a large error compared with other characteristic bands, which may be caused by the uneven layer of wheat leaves during the measurements. The higher CV of band 1 (Red) is probably due to the small mean reflectance.

### 4.2. An Acceptable Error from the Point of View of Reflectance

If the measured reflectance values that distinguish FVC objects are stable over time, then the derived spectral indices established by the characteristic bands should also be stable over that time, thus, the FVC values estimated were assumed to be stable. In this study, the reflectance values of the objects fluctuate more over time than the vegetation indices do. The vegetation indices removed the variation in the original reflectance values to some extent due to the ratio operation.

Duggin suggested that if the error of reflectance coefficient is 10% [22,34], the measurement results were acceptable when they analyzed the difference of surface reflectance under different irradiance conditions on sunny and cloudy days. In our experiment, the CV of reflectance in characteristic spectral bands in the period of 10:00 a.m.–16:00 p.m. were less than or about 10%, and the CV of the 36 repetitions at each time point during the period 10:00 a.m.–16:00 p.m. was approximately 10%, so it was consistent with the previous results. The CV of the spectral indices derived from the characteristic bands in our study was stable between 9:00 a.m. and 17:00 p.m., and the ANOVA test further confirmed this result.

### 4.3. Limitations of the Experiment

From the above results, it is learned that when field spectral measurements are made to verify FVC estimation by remote sensing, measurements can be scheduled one and a half hours after sunrise and one and a half hours before sunset on a clear winter day. This was a broader time frame than other studies and national standards have suggested. For example, when Cao and Wang collected spectra in the field to distinguish FVC objects, there were only four hours between 10:00 a.m. and 14:00 p.m. [13,35], the national standard for measuring spectra of objects is between 10:00 a.m. and 15:00 p.m., leaving only five hours for fieldwork [11]. Even the longest time period is only 6 h from 10:00 a.m. to 16:00 p.m. in Guerschman's field experiment [12]. In this study, however, based on the analysis of the stability of the spectral indices over time, the recommended total of 8 h between

9:00 a.m. and 17:00 p.m. is longer than others' sampling schedules. Collection of field spectra at the recommended wider time frame made it feasible and easier for field spectral library and further image analysis and spectral index development.

It is worth noting that the experiment in this study was basically carried out near the time of the lowest solar altitude angle in the northern hemisphere (22th December), which could provide reference for the area above 34 degrees north latitude. Readers must check the data in this study and understand the limitations of the results in this experiment. It is reasonable to believe that measurements carried out when the solar altitude angle is higher (e.g., summer) are expected to return even better results.

## 5. Conclusions

In order to determine and assess the appropriate time for FVC estimation using satellite remote sensing, an innovative field experiment was carried out to continuously measure PV, NPV and BS targets over a three-day period. The spectral curves were measured between 7:30 a.m. and 17:30 p.m. for five objects using a spectrometer; the variation of reflectance and FVC spectral indices over time was analyzed and the following conclusions were obtained.

- NDVI and SWIR32 can potentially apply in distinguishing PV, NPV and BS objects.
- The degree of stability of the reflectivity over time varies for different FVC objects and bands. Generally, the appropriate time to obtain the relative stable reflectance is from 10:00 a.m. to 16:00 p.m. with the CVs for different bands ranging from 5.01% to 9.53%.
- The appropriate measurement time to obtain FVC indices (NDVI and SWIR32) varies for the nature of objects. The time for PV was 7:30 a.m.–17:30 p.m., with a CV of 0.75% and for NPV1–3 and BS it was 9:00 a.m.–17:00 p.m., with CV ranging from 2.09% to 3.1%.
- Though the reflectivity of the characteristic bands is varied and scattered, the derived spectral indices are more stable over the measuring time. An extended period (9:00 a.m.–17:00 p.m.) might be acceptable depending on the variation of the spectral index values over time.

**Author Contributions:** Methodology, B.L., D.L. and X.Z.; software, D.L., L.H.; investigation, D.L., L.H., J.H., J.G., J.W., B.L., X.Z., Q.C.; formal analysis, B.L., D.L., X.Z.; data curation, D.L.; writing-original draft preparation, D.L.; writing-review and editing, D.L., B.L., X.Z., X.Y.; project administration, B.L., X.Z.; funding acquisition, B.L., X.Z.; resource, B.L., X.Z. All authors have read and agreed to the published version of the manuscript.

**Funding:** This research was jointly supported by the Strategic Priority Research Program of Chinese Academy of Sciences (Grant No. XDA 20040202), and National Natural Science Foundation of China (Grant No. 41877083).

**Acknowledgments:** We are very grateful to Qingrui Chang from the Northwest Agriculture and Forestry University, Yangling, China, for providing us with the necessary equipment and training in the operation of the experiment.

**Conflicts of Interest:** The authors declare no conflict of interest.

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
