# Peer review of "An Experimental Study on Field Spectral Measurements to Determine Appropriate Daily Time for Distinguishing Fractional Vegetation Cover"

_remotesensing, doi:10.3390/rs12182942_

Round 1

Reviewer 1 Report

Review report on manuscript number ID remote sensing-906065 submitted to "remote sensing".

Title: An experimental study on field spectral measurements to determine appropriate time for distinguishing fractional vegetation cover

Authors: Du Lyu, Baoyuan Liu, Xiaoping Zhang, Xihua Yang, Liang He, Jie He, Jinwei Guo and Jufeng Wang

This study manuscript the authors measured five groundcover objects from 7:30 to 17:30 local time in three consecutive sunny days using a portable spectrometer. They analyzed the spectral curves among these objects and the variation in corresponding to the MODIS satellite channels. Τhe subject is interesting and relevant to the field of this journal. On the other hand, the paper has some key omissions that have to be corrected before it is suitable for publication.

The language should be improved (Moderate English changes required).

The abstract does not provide the reader with information about the results obtained. It has not any numerical values. It needs to be improved, giving more numerical values for the results.

The authors should change all the sentences in the text which are written in the first plural (line 20, 21, 23, 47, 55, 83, 88, 146, 162, 163, 192, 204, 220, 259, 294 and 307). 

The authors should compare their work with other works. Did the authors try to compare their work with other works? 

Avoid lumping references ([4-7] and [10, 15–17]). Instead, summarize the main contribution of each referenced paper in a separate sentence and/or cite the most recent and/or relevant one. The authors should also consider adding there recently published results in the field.

In Section 3 the authors present the experimental results. The results are not presented in an understandable way. This section should be rewritten again more analytically. 

The Conclusion is not suitable, should give more useful conclusions. This section is too poor and presented in brief. This paragraph should be reformed. Should include numerical values for the results.

Reviewer 2 Report

Dear Authors,

Thank you for considering Remote Sensing for your interesting work. Despite its overall potential and scientific soundness I see some points which must be strengthened in your revised version of the manuscript. 

1) the spectral measurements are carried out in consecutive days but only in December - why? 2) the solar zenith angle changes from a day to day not so much - so it can be neglected to a degree - however, how do you account for the seasonal changes in the illumination conditions? 3) the spectral data is compared with MODIS spectral indices. How do you make sure that the pixels of MODIS - even they are mixed ones - are correctly geolocated? Why don't you use drone or VHR data instead? 4) narrow band vegetation indices are compared with broadband vegetation indices from MODIS - is this so? 5) Could you carry out these experiments under controlled conditions to check for all the parameters under scrutiny? 6) to what extent the tested vegetation indices can be taken as a proxy to the FVC? If they are not equate to the FVC why do you test FVC estimation while you test the vegetation indices changes over time? Where is here the novelty as it is a well known fact of the directional changes of the spectral measurements due to many factors. 7) What is the set-up for spectral measurements and what is the satellite view angle? How do you account for the discrepancies of the two in your work? 8) Why don't you compare your field measured data with simulated spectra? 9) Do you account for the uncertainties - for the the satellite estimates and field spectra derived vegetation indices? 10) Last but not least - how do you measure FVC on the field?

These are just some of the many questions related with the study methodology which I would like to see answered in your revised manuscript.

Kind regards,

Reviewer 

Reviewer 3 Report

See attached file

Round 2

Reviewer 1 Report

All the comments have been taken seriously into account.

Reviewer 2 Report

Dear Authors,

Thank you for your detailed answers to my remarks. Indeed you managed to answer well. I slightly disagree with some of the answers but I agree that the study is limited to an extent which you do not want to expand. This is the reason why accept all of your rebates which allows me to accept the work as a whole and recommend it to the editor. I wish you all the best with your future studies and also to focus more on the links with remote sensing which is the end-user of this work. 

Kind regards,

Reviewer

Reviewer 3 Report

Accepted